# Damping Behavior of Hybrid Composite Structures by Aeronautical Technologies

**Alice Proietti [1], Nicola Gallo [1], Denise Bellisario [2,*], Fabrizio Quadrini [1] and Loredana Santo [1]**

1. Department of Industrial Engineering, University of Rome 'Tor Vergata', via del Politecnico 1, 00133 Rome, Italy
2. Faculty of Ecomics, Universitas Mercatorum, Piazza Mattei 10, 00186 Rome, Italy
* Correspondence: denise.bellisario@unimercatorum.it

**Abstract:** Hybrid composite laminates are manufactured by using technologies and raw materials of the aeronautic sector with the aim to improve the damping behavior of composite structures. Matrix hybridization was achieved by laminating carbon fiber reinforced (CFR) plies with elastomer interlayers. Up to 10 different composite sandwich architectures were investigated by changing the stacking sequence, the thickness of the elastomer layers, and the elastomer typology, whereas the total number of the CFR plies was fixed to six for all the hybrid composites. Square panels with the size of $300 \times 300$ mm$^2$ were autoclave molded with vacuum bagging, and rectangular samples were extracted for static and dynamic tests. Dynamic mechanical analyses were performed to measure the storage modulus and loss factor of hybrid materials, which were compared with static and dynamic performances of the composite structures under bending. Repeated loading–unloading cycles and free oscillation tests allowed us to the energy loss per unit of volume, and the acceleration damping, respectively. Results show that softest elastomer interlayers lead to big loss of stiffness without any positive effect in the damping behavior, which worsens as well. By using soft elastomers, complex architectures do not provide any additional benefit in comparison with the traditional sandwich structure with soft core and hard skins.

**Keywords:** hybrid composite materials; carbon fiber composites; damping; autoclave molding

## 1. Introduction

The term "hybrid", from Latin "hybrida", has been used, for long time, to describe the offspring of two plants or animals of different species or varieties. Generally, it refers to a thing made by combining at least two different elements. The hybrid concept, applied to materials, refers to when the combination of these different elements allows for the production of some new and unexpected properties and functions. In the case of composites, the term hybrid refers to the composite material which is composed of either one fiber material along with one or more matrix materials, or one matrix material along with one or more fiber materials [1]. Most of the proposed cases for the aerospace sector refer to the use of different fibers (organic, carbon, natural or glass fibers) in the same organic matrix. In typical micro-composite materials, the mixture of matrix and fiber results in averaging the properties of both, whereas new properties have to be achieved by hybrid composites [2]. For example, vehicles and packaging products have proved that natural fibers are a viable alternative to synthetic fibers. In fact, hybridization offers additional advantages since the mixing of cheaper, low-quality fibers with more expensive fibers of higher quality can improve the properties of a composite without significantly affecting the cost [3]. Among the improvements which resulted from hybridization are those affecting the damping or the fatigue behavior. All these aspects make hybrid composites very attractive for high-added value markets with severe technical constraints, such as the automotive and transportation, marine, aerospace, wind energy, and sports markets [1].



The material class of hybrid composites is very large, and research studies are uncountable. Approximately 40,000 manuscripts refer to "hybrid composite materials" in the main relevant fields (title, abstract, keywords); more than 2000 already in the current year (Scopus font). Nevertheless, research trends are very clear, and some important achievements have been reached recently.

In the last year of research, by restricting the analysis to structural composites, the most common form of hybridization refers to the contemporary use of carbon and glass fibers. These fibers are generally inserted in an epoxy matrix, and conventional manufacturing procedures are adopted [4]. In terms of the expected improvement from hybridization, most of these studies discuss positive effects for the dynamic behavior of hybrid materials, improving qualities such as fatigue [4,5], impact [6,7], and damping, with and without aging [8,9]. However, they continue to propose the typical composite hybridization by combing carbon and glass fibers, mainly into an epoxy matrix. In fact, carbon fiber reinforced (CFR) and glass fiber (GF) composites are the most used composite materials for transportation, but they are rarely combined in the same product because of the very different static performances of glass and carbon fibers. In some cases, instead of damping properties, which are measured under cycling, effects on toughness are investigated, typically extracted from quasi-static tests [10]. Results confirm that the hybrid composite shows lower stiffness and strength than traditional CF laminates, but it has the advantage of an increased damping behavior. Hybridizing CFR laminate with GF lamina is also found to be an efficient way to improve delamination tolerance without reducing laminate resistance to flexural and impact loads [11]. Toughness improvements are also found by combining CFs with other fiber types, such as polyarylate and polybenzobisoxazole [12], Kevlar [13,14], aramid [15], liquid-crystal polymers [16], basalt [17], and carbon nanotubes [18–20]. The interaction mechanisms seem to be the same as the common CF–GF hybrid composites, but different extents of damping properties may be reached by tuning the related compliance.

The hybridization of thermoplastic matrices was also investigated, showing positive interactions in static and dynamic behavior both in the case of glass and carbon fibers with polypropylene [21], as well as for carbon fibers and metallic braided wire mesh with poly-ether-ether-ketone [22]. In this case, an important contribution to damping is made by the matrix, too, but problems with optimal matrix–fiber adhesion may arise for such non-traditional fibers. In other cases, hybridization is used as a tool of sustainable development by integrating traditional fibers with natural ones, such as jute [23–25], flax [26], kenaf [27], banana fibers [11], cellulose [15], or sisal [28]. For these studies, the aspect of aging is most significant because of the strong sensitivity of natural fiber to the environment. As a consequence, final composite performances are often low, and not attractive for highly demanding application fields.

Hybridization is also exploited in terms of structures if the interface between the different elements occurs at the macro-scale. Examples are the hybrid panels of 7075-T6 aluminum alloy and CFR laminate, manufactured using the thermoforming process [29], pultruded rods [30], hybrid joints [31], and sandwiches [32]. These structures are easier to implement into the design and manufacturing procedures of the aerospace sector, but attention has to be paid at the interface in order to detect unexpected adhesion or corrosion problems. Dealing with structures, hybrid assemblies can be produced by joining thermoplastic (TP) and thermosetting (TS) matrices [33]. Co-bonding generally occurs during composite laminate molding. Nevertheless, co-bonding of TP and TS matrices remains as a challenge in the manufacturing of hybrid composites [34]. During curing, TP films, in contact with TS layers, can partially penetrate, leading to macro-mechanical interlocking as the main connection mechanism [35]. These hybrid composites, also known as "multi-material composites", combine the damage tolerance characteristics of thermoplastics with the strength and stiffness of TS composites, resulting in more reliable structures with an extended lifetime [36]. In fact, TP matrices are more ductile than TS, leading to higher damping, toughness and impact resistance at low speed. The same hybrid effect of TS–TP systems is proposed in this study by combining an elastomeric matrix with the traditional

epoxy matrix of CFR prepregs. The use of a thermosetting elastomer is expected to improve co-bonding with the prepreg epoxy matrix. Moreover, commercial raw materials and conventional molding procedures used in aerospace have been used to manufacture the samples. This study aims to show that hybrid elastomer-rigid CFR TS laminates can be produced by using already-available materials and equipment. For the first time, hybrid composite laminates have been manufactured by interposing uncured rubber interlayers between CFR thermosetting prepregs during the lamination procedure. Another novel aspect is that hybrid composites have been produced by using state-of-the-art aeronautical materials and molding technologies.

This study has been carried out to develop new smart structures for enhanced vibration mitigation. In fact, aeronautic structures experience vibration and impulsive loads that can be propagated and amplified, thus reducing their lifespan. Structural dampers are used to address this issue, but smart structures could integrate this capability into the material. However, very stiff structures behave poorly under dynamic loads, and a compromise is necessary between static and dynamic performances. The insertion of viscoelastic interlayers between composite plies during the lamination process is a smart manufacturing solution for these multi-functional structures [37]. Lightweight composite sandwich beams have been manufactured using this technique. The elastomeric sheet has the additional role of mitigating the thermal expansion mismatch between composite plies and aluminum sheets [38]. Galvanic protection at the interface is also provided. By optimal design of the elastomeric layer, an increase in damping of approximately 23% has been achieved. CFR plies are also used in these smart structures [39]. Damping improvements from 25% to 45% have been already found.

The article is organized into five sections as follows: in Section 2, material properties, sample architectures and manufacturing, as well as the whole experimental procedure are reported. The results of both static and dynamic tests are shown in Section 3 and then fully discussed in Section 4. The final section includes a summary of the paper's main results and the conclusions.

## 2. Materials and Methods

The full experimental procedure is shown in Figure 1. Hybrid composites have been manufactured in autoclave by using commercial CFR epoxy TS prepregs (Solvay Cycom 977-2, woven fabric). Elastomeric layers were made by using 2 different rubber materials which are commercially applied as auxiliary materials during composite manufacturing, and are supplied in the shape of uncured sheets.

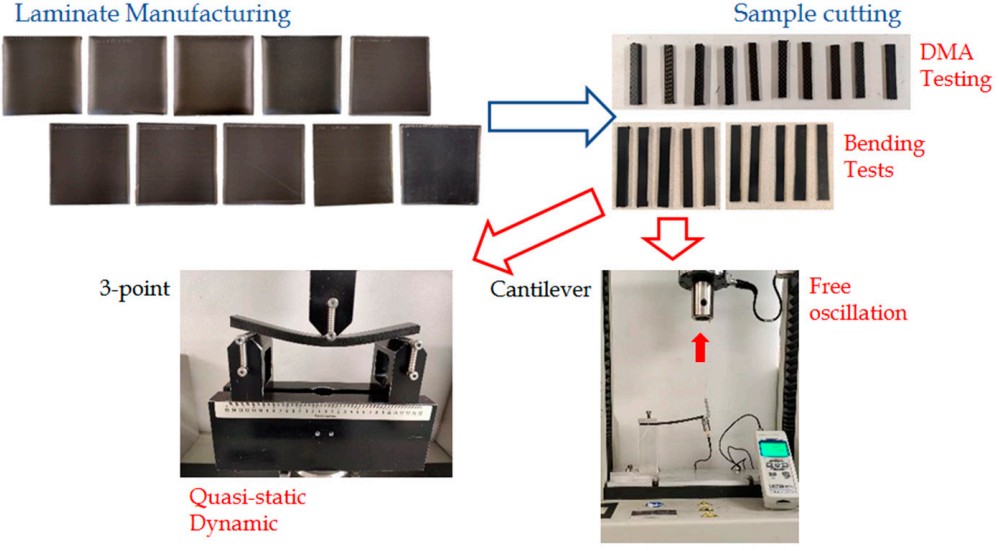

**Figure 1.** Experimental methodology.

The first (AirPad by AirTech) is an uncured, non-silicone rubber for the manufacture of caul sheets, flexible mandrels, and rubber tooling. It works as a pressure intensifier during autoclave processing and improves component quality from the vacuum bed side of the part. It was supplied in the form of an uncured sheet, on a roll, with an initial thickness of 1.59 mm. The second elastomer (AirTech Pressure Strip) is an uncured modified butyl rubber to be placed in corners where pressure is difficult to apply with only a vacuum bag. This second elastomer was also supplied in rolls, but with an initial thickness of 3.18 mm. In composite manufacturing, these elastomers are used as expendable items and discarded after each cure. In this study, for the first time, they are used to produce co-cured hybrid CFR laminates. In the following, they are referred as rigid (AirPad) and soft (AirTech) elastomers because of their different rigidity.

### 2.1. Laminate Architecture

Different architectures have been considered for the hybrid composite laminates. The common feature of all these architectures is that 6 CFR plies have always been used. Differences were made in terms of type of elastomer (rigid and soft), stacking sequence, and number of rubber plies, according to Figure 2. The first 4 architectures of the hybrid composite panels were used in both elastomers, whereas the last 2 were used only in the rigid elastomer. Labels of the structures report the stacking sequence during lamination, as in Table 1.

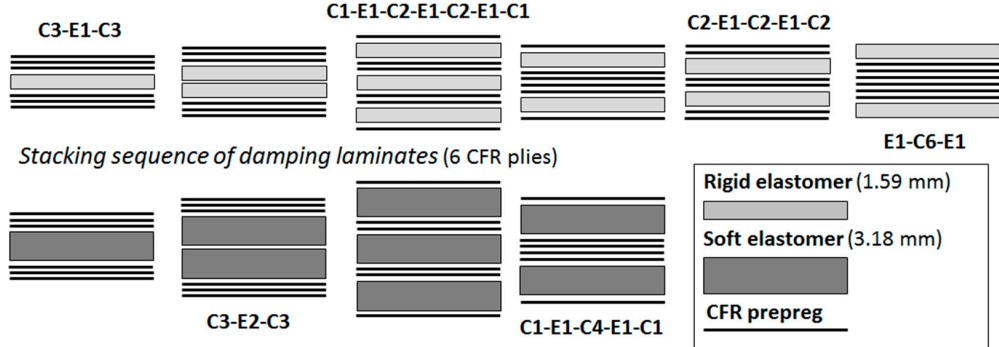

**Figure 2.** Architectures of the hybrid composite structures (C = CFR ply, E = elastomer layer).

**Table 1.** Stacking sequence of the molded plates.

| Label | Stacking Sequence | Elastomer |
|---|---|---|
| C3-E1-C3 | C/C/C/E/C/C/C | Rigid, Soft |
| C3-E2-C3 | C/C/C/E/E/C/C/C | Rigid, Soft |
| C1-E1-C2-E1-C2-E1-C1 | C/E/C/C/E/C/C/E/C | Rigid, Soft |
| C1-E1-C4-E1-C1 | C/E/C/C/C/C/E/C | Rigid, Soft |
| C2-E1-C2-E1-C2 | C/C/E/C/C/E/C/C | Rigid |
| E1-C6-E1 | E/C/C/C/C/C/C/E | Rigid |

In composite sandwiches, rigid plies are incorporated into the external skins to increase the bending stiffness of the structure. In fact, composite laminae, close to the neutral axis, provide a small contribution to the overall stiffness. Nevertheless, soft sandwich cores reduce their thickness in the loading zone, thus reducing the distance between the external plies. If this effect is relevant, the presence of rigid CFR plies, close to the sample axis, can be more beneficial to the bending stiffness than in the external skins. The proposed architectures aimed to experimentally measure this transition.

### 2.2. Autoclave Molding

The production of the hybrid composites has been carried out by using state-of-the-art materials, equipment, and procedures of the aeronautic sector. The CFR plies and rubber

layers were cut in the shape of $300 \times 300$ mm$^2$ square sheets, and laminated according to the architectures of Figure 2. A vacuum bag was used, and all 10 plates cured contemporarily in autoclave for 3 h at the temperature of 180 °C at the pressure of 6 bar, according to the datasheets of the raw material manufacturers, with a heating and cooling rate of 2 °C/min. The measured cured ply prepreg thickness was 0.24 mm and the measured areal weight of the prepreg was 335 g/m$^2$. During the cure, CFR plies and elastomer layers consolidated and co-bonding at the CFR/elastomer interface occurred. After molding, rectangular samples were cut for testing from all the hybrid laminates, far from the edges.

Average thickness and density of molded plates are reported in Table 2. Sandwich densities are lower than conventional CFR laminates with epoxy matrix, ranging between approximately 1.4–1.5 g/cm$^3$, because of the lower density of the elastomer layers. In fact, by reducing the number of elastomer layers, the density reduces. For the same reason, by increasing the thickness, the density reduces as well. By comparing hybrid laminates with rigid and soft elastomer layers, the thickness of the soft class is 60% thicker than the rigid, on average, because of the initial thicker uncured plate.

**Table 2.** Thickness and density of molded hybrid composite laminates.

| Hybrid Laminate | Thickness, mm | Density, g/cm$^3$ |
|---|---|---|
| *Rigid elastomer* | | |
| C3-E1-C3 | 2.88 | 1.26 |
| C3-E2-C3 | 4.79 | 1.23 |
| C1-E1-C2-E1-C2-E1-C1 | 5.86 | 1.15 |
| C1-E1-C4-E1-C1 | 4.35 | 1.19 |
| C2-E1-C2-E1-C2 | 4.36 | 1.18 |
| E1-C6-E1 | 4.35 | 1.18 |
| *Soft elastomer* | | |
| C3-E1-C3 | 4.28 | 1.28 |
| C3-E2-C3 | 7.87 | 1.20 |
| C1-E1-C2-E1-C2-E1-C1 | 9.89 | 1.20 |
| C1-E1-C4-E1-C1 | 4 | 1.28 |

Density and thickness are important parameters to determine the dynamic behavior of the hybrid laminates, as natural frequencies increase by reducing the density, and increase by increasing the thickness.

The appearance of the cross-section of hybrid plates is shown in Figure 3 by Leica S9i stereoscope. Magnifications of some interfaces are proposed. As expected, a very good adhesion was found between adjacent rubber layers and adjacent CFR plies. Moreover, the same optimal adhesion was observed at the hybrid interface between elastomers and CFR plies. Co-bonding was very effective thanks to the common thermosetting nature and the chemical affinity of the components. Moreover, the rubber resin bleeding, and the similar processing window for curing, aimed for this high adhesion.

### 2.3. Testing Procedure

Rectangular samples with the size of $60 \times 10$ mm$^2$ were extracted by the hybrid laminates for dynamic mechanical analysis (DMA), one sample from each laminate. Isothermal tests (by Netzsch DMA 242C) were performed in 3-point bending configuration for 5 min at room temperature (25 °C) with the frequency of 10 Hz and the span-length of 40 mm. Neat samples of elastomers (soft and rigid) and CFR laminate (3-ply) were extracted for comparison from the hybrid composites The final value of storage modulus (E') and loss factor (tg δ) were used for comparison.

After DMA testing, the same hybrid specimens were used for quasi-static bending tests by the universal material testing machine (Insight 5 by MTS). The same 3-point bending configuration of DMA was adopted, with the span-length of 40 mm, at the rate of 1 mm/min up to 1 mm of maximum displacement and at room temperature (25 °C, 40% RH). The elastic modulus (E) was extracted from quasi-static bending tests to be compared

with E'. In fact, differences are expected because of the effect of the loading conditions (cyclic and quasi-static) and the adopted elasticity range. DMA tests analyze samples in a micro-range (up to 120 μm), whereas quasi-static bending tests use a large displacement range (up to 1 mm). Rubbers and elastomers are particularly sensitive to the elastic range of testing, being more rigid at low displacements.

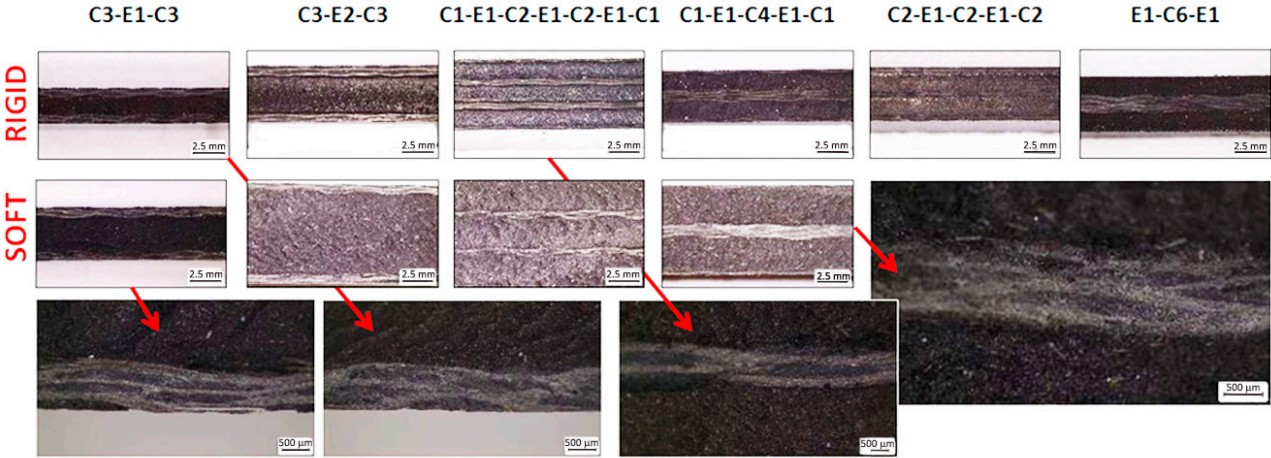

**Figure 3.** Cross-section of hybrid laminates and details of such interfaces.

For dynamic testing in the universal material testing machine, larger size samples were extracted (150 × 30 mm²), 4 samples from each laminate. One of these samples, for each type, was used for cyclic bending tests. In this dynamic test, 10 loading–unloading cycles were repeated by applying a maximum displacement of 10 mm at the testing rate of 10 mm/min, with an initial pre-load of 0.5 N and a span length of 120 mm at room temperature (25 °C, 40% RH). The testing setup details are depicted in Figure 4a, for the quasi-static test. Inversion occurred at the maximum displacement and at the pre-load. Figure 5 shows a typical cyclic bending test in terms of load-time and stress–strain curves.

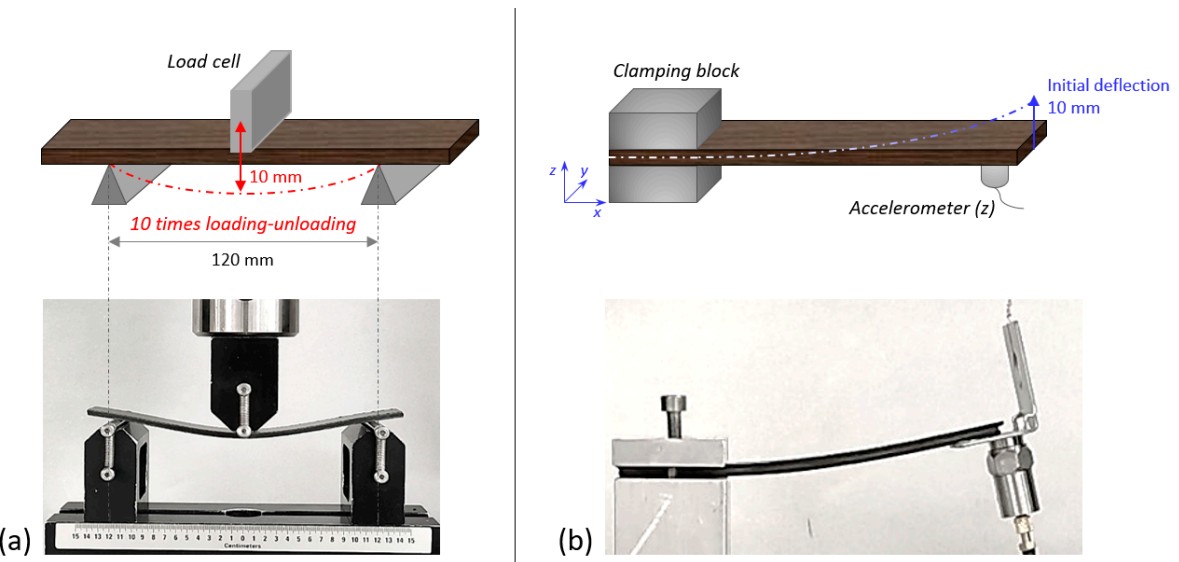

**Figure 4.** Testing setup details for quasi-static tests (**a**) and for dynamic free oscillation tests (**b**).

A small decrease in the load peak is observed during cycling (Figure 5a), primarily after the first cycle, and rarely in the last cycles. The stress–strain curve (Figure 5b) shows the typical hysteresis of damping, the area of which is the evaluation of the energy loss in that loading–unloading cycle.

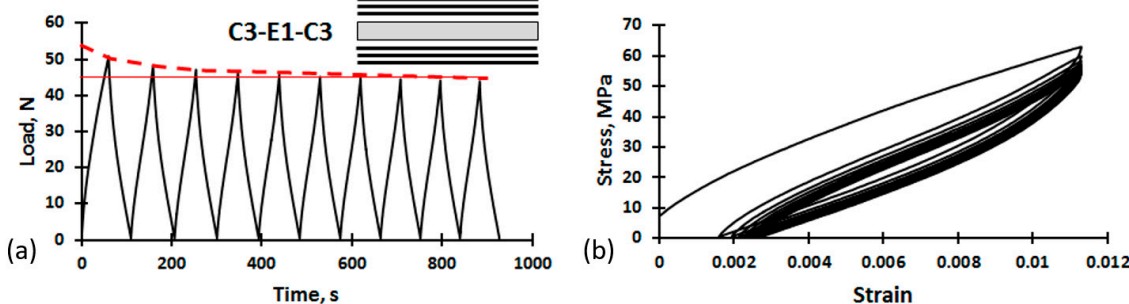

**Figure 5.** Results from cyclic bending tests in terms of load-time (**a**) and stress–strain curves (**b**).

The last test was performed to evaluate the behavior of the hybrid samples under free oscillations. Single cantilever configuration was used considering a span length of 120 mm and the maximum displacement of 30 mm. The tests were conducted at room temperature (25 °C, 40% RH). The free end was fixed by a metal wire to the testing machine cross-head to apply the initial inflection. An accelerometer was placed on the bottom surface of the sample at the same end. By cutting the wire, it was possible to measure the acceleration damping over time. As shown in Figure 4b, acceleration data were acquired by using a mono-axial accelerometer fixed on the free end. A vibration meter PCE-VT-2800, with resolution of 0.1 m/s$^2$ and accuracy of $\pm$ 5%, was used with the acquisition rate at 160 Hz.

## 3. Results

The damping behavior has been measured by DMA, free oscillation and cycling testing. Results are expressed in terms of loss factor (which is close to half of the material damping coefficient), energy spent per loading–unloading cycle, and acceleration reduction under free oscillation. These data provide a description of the material damping capabilities under a wide spectrum, from small to large displacements. They are not dependent on one another but a correlation may exist. Generally, an improvement of the damping behavior of a structure leads to a general increase in all these parameters, but to different levels of extent.

Results from DMA and quasi-static bending tests on small specimens are reported in Table 3. A strong correlation is present between the storage modulus and the quasi-static modulus, as shown in Figure 6a. The quasi-static modulus is, on average, 40% of the storage modulus, with a very good approximation.

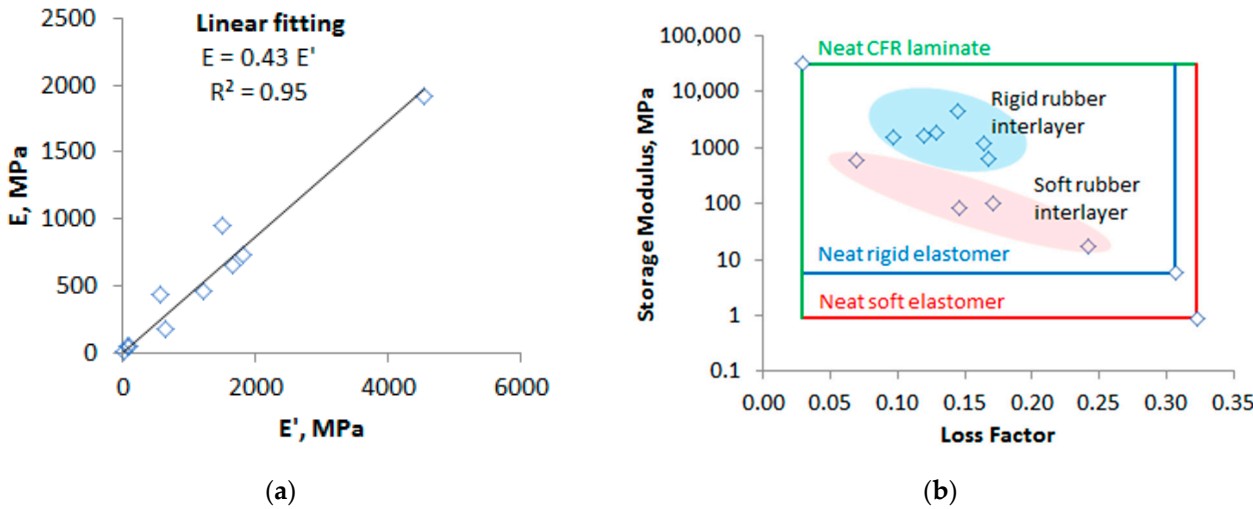

(**a**)

(**b**)

**Figure 6.** Correlation of the storage elastic modulus E′ with the quasi-static modulus E (**a**), and the loss factor tg δ (**b**).

**Table 3.** Data from DMA and quasi-static bending tests on small samples.

| Hybrid Laminate | E', MPa | tg δ | Stiffness, N/mm | E, MPa | Loss Energy, J/cm³ | Acceleration Damping, % |
|---|---|---|---|---|---|---|
| *Rigid elastomer* | | | | | | |
| C3-E1-C3 | 4546 | 0.145 | 24.0 | 1919 | 63.3 | 90.9% |
| C3-E2-C3 | 1662 | 0.120 | 34.2 | 659 | 72.3 | 92.5% |
| C1-E1-C2-E1-C2-E1-C1 | 649 | 0.167 | 18.3 | 175 | 52.3 | 91.7% |
| C1-E1-C4-E1-C1 | 1830 | 0.129 | 32.6 | 734 | 61.4 | 89.1% |
| C2-E1-C2-E1-C2 | 1213 | 0.164 | 19.2 | 453 | 53.9 | 88.9% |
| E1-C6-E1 | 1498 | 0.097 | 39.1 | 956 | 12.8 | 93.9% |
| *Soft elastomer* | | | | | | |
| C3-E1-C3 | 574 | 0.070 | 16.1 | 428 | 7.2 | 90.2% |
| C3-E2-C3 | 83 | 0.145 | 11.1 | 48 | 5.8 | 88.5% |
| C1-E1-C2-E1-C2-E1-C1 | 17 | 0.242 | 3.7 | 8 | 4.1 | 80.0% |
| C1-E1-C4-E1-C1 | 98 | 0.171 | 8.5 | 51 | 7.2 | 88.9% |
| **Neat material** | | | | | | |
| 3-ply CFR laminate | 31,625 | 0.029 | | | | |
| Rigid elastomer | 6 | 0.307 | | | | |
| Soft elastomer | 1 | 0.323 | | | | |

The large distance between E and E' confirms the rubber-like behavior of the hybrid composites. Moreover, their strong linear correlation gives validity to the adopted DMA testing procedure.

Results from damping tests on hybrid samples are also reported in Table 3, in terms of average hysteresis area (loss energy) under cyclic bending, and acceleration reduction during free oscillations. The hysteresis was averaged in the last five cycles where the peak load reduction was negligible, thus reducing the standard deviation under 5% for all the tests. The acceleration reduction was calculated in the first 5 s of oscillation. All the hybrid samples damped the full oscillation in approximately 10 s, thus the evaluation was made in the middle of this time range. Data from acceleration damping are expressed as the percentage decrease from the initial value and were averaged on the three tested samples.

## 4. Discussion

The loss factor is representative of the intrinsic damping behavior of a material. In a rough approximation, it is half of the damping coefficient. Materials with high loss factors generally show a loss of storage modulus. Therefore, it is always necessary to find the optimal compromise between the stiffness and damping of structures, depending on the design requirements. In complex structures, such as the hybrid composite sandwiches, the damping and elastic properties of the elements are averaged. In Figure 6b, the storage modulus is reported as a function of the loss factor for all the samples. As a general trend, E' decreases by increasing tg δ, but some exceptions are present. Hybrid composites, with rigid rubber interlayers, have a higher storage modulus than those with soft interlayers, independently from the architecture. Moreover, all the hybrid composites have a storage modulus lower than the neat CFR laminate, but higher than the neat elastomers. Conversely, their loss factor is always higher than that of the neat CFR laminate, but lower than that of neat elastomers. As a consequence, hybrid laminates have dynamic properties in the middle of the neat samples, and the type of the elastomer interlayer is recognizable as well.

Elastomers have a very low storage modulus and very high loss factor in comparison with the neat CFR laminate: E' is four orders of magnitude higher, tg δ is one order of magnitude lower. In particular, the loss factor of the soft elastomer is 0.32, and 0.5 is considered as a reference for optimal damping. This large mismatch between the sandwich components produces low rigidity panels. For both elastomers, moving from the structure with one internal rubber layer (C3-E1-C3) to two (C3-E2-C3), the stiffness reduces: 63% for the rigid elastomer and 86% for the soft elastomer. The stiffening effect of the higher distance of the composite skin is already lost by doubling the rubber core. Other data

also confirm that all the hybrid composites are dominated by the elastomer compliance. Lower elastomer thickness would lead to improvement in the composite stiffness, but this goal requires different raw materials which are still not available. The hybrid panel with all the six CFR plies together (E1-C6-E1) shows the highest stiffness, 63% higher than C3-E1-C3, but the loss factor is very close to the neat CFR laminate. Therefore, in absence of an elastomer interlayer, the composite panel has almost no damping.

An evident result is that all the hybrid composite laminates have very good agglomeration despite the different mechanical behaviors of CFR plies and elastomers. The feasibility of manufacturing this new typology of hybrid composite has been shown by using the state-of-the-art technologies of aeronautics, even if the optimal thickness of the elastomer layer has yet to be found. The relationship between the dynamic behavior of the composite structure and the properties of the single components is very complex because of the deformation mechanisms which are involved during loading. An example is given by the simplest architecture (C3-E1-C3) where the elastic modulus of the hybrid panel with soft elastomer is lower than that of the rigid, but its loss factor is much higher, not far from the neat CFR laminate. This architecture seems to be very bad for the soft elastomer as the very low modulus is associated with a very low loss factor. The poor damping behavior in comparison with the other hybrid composites is also confirmed by the other damping properties, energy loss and acceleration damping of Table 3.

In general, hybrid composites with the soft elastomer do not show better damping properties of those with the rigid elastomer. By comparing the same stacking sequence, apart from the discussed C3-E1-C3 case, their loss factor is higher, but energy loss and acceleration damping are lower. This strong mismatch may be justified by the effect of the displacement range, moving from 100 μm of DMA tests to 10 mm of dynamic bending tests. The low rigidity of the soft rubber layer is also negative for the damping as the material is not able to consume the acquired energy. As an average on the four hybrid samples with a comparable architecture, hybrid composite laminates with the soft elastomer lose 5% of the acceleration damping. This occurrence suggests that the properties of the soft elastomer are not appropriate for a good hybrid structure as the loss in stiffness is not compensated by an improved damping. A threshold for the rubber compliance is present, over which the deforming mechanisms of the hybrid structures do not allow for further damping improvement. Moreover, the displacement scale of these deforming mechanisms is important and data from DMA tests are not sufficient to predict the behavior of complex structures under loading.

It is difficult to make a comparison between the proposed architectures as it is not possible to state the acceptable reduction in stiffness for a given increase in damping. An experimental approach may be combining the measured data for stiffness and damping. In particular, the specific modulus is considered, i.e., the ratio of the elastic modulus of Table 3 with the density of Table 2. High values of the specific modulus lead to high stiffness per unit of weight, which is fundamental in aeronautics. Moreover, natural frequencies are related to the square root of the specific modulus. The specific modulus is reported in Figure 7 for all the hybrid composites, together with the data of loss energy and acceleration reduction in Table 3. The data are normalized, by the minimum and the maximum values, to obtain 1 for the best and 0 for the worst case. This is confirmed by the fact that the choice of the soft rubber is negative for all the architectures. The most complex configuration C1-E1-C2-E1-C2-E1-C1 shows the lowest values of all the parameters. Instead, the best combination of properties is found in the simplest structure C3-E1-C3 with rigid elastomer core. It is also confirmed that rubber pads on the top and the bottom of the laminate (E1-C6-E1) do not provide good dynamic performances.

The use of an interlayer is suggested but the thickness of this interlayer should be designed on the basis of the properties of the elastomer. In this study, manufacturing constraints have forced us to use thick layers of elastomers, and this solution seems to affect the final properties of the hybrid laminates. In the case of low modulus rubbers, the use of complex architectures does not enable us to achieve good dynamic behaviors. In

fact, Figure 7 shows that in the top of the list there are two laminates with the traditional architecture of soft core and hard skins.

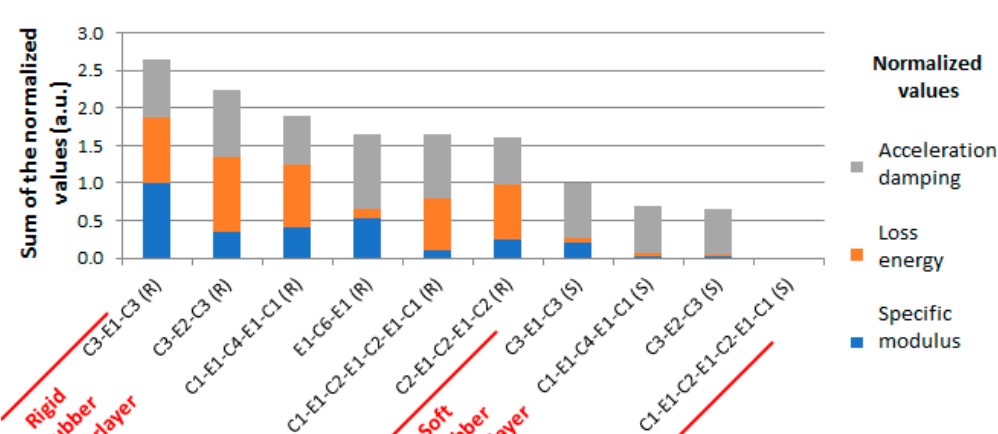

**Figure 7.** Normalized static and dynamic performances of all the hybrid composites.

The proposed architectures and material combinations are quite new in the scientific database; therefore, a direct comparison with other similar studies is difficult to perform. Nevertheless, the known advantage in terms of mixing the elastic and damping properties of the raw materials has been observed, even if the final products exceed the damping performances, thus strongly reducing the elastic part.

## 5. Conclusions

Hybridization is a tool of material design to obtain unexpected properties from structures by combining the performances of very different components. Matrix hybridization seems to be a very effective way to incorporate damping behavior into stiff composites, despite the issue of a good interface between hard composite plies and soft interlayers. The main achievement of this study is that hybrid composite laminates, with elastomer-epoxy matrix hybridization, can be manufactured by using commercial materials of the aeronautic market. However, elastomer interlayers were made using uncured rubbers used for different aims such as pressure intensifiers or caul sheets. Their technological and material properties are not optimized for hybrid composite production, mainly the initial thickness of the uncured sheet. However, important indications can be deduced from the experimentation with the different architectures. The use of the softest rubber affects the damping behavior of the structures as well as their stiffness. Simple architectures behave better than complex, if the thickness of the elastomer layer is large. Nevertheless, all the hybrid composites showed very good adhesion after co-curing, despite the different nature of the prepreg matrix and the uncured elastomers. The best hybrid laminate already shows interesting properties, such as a modulus higher than 4.5 GPa (14% of the neat CFR laminate) with a damping coefficient one order of magnitude higher than the sole CFR laminate. In the future, new architectures will be tested by taking into consideration different configurations, such as free samples under impact probes.

This study shows that producing smart structures for aeronautics is possible by using state-of-the-art materials and procedures. In order to enhance vibration mitigation, the applied solution involves interposing elastomeric interlayers during lamination [38,39]. Final samples exceed in the damping behavior, but these performances can be tuned, in the future, by using different architectures or layer thicknesses. At present, the developed materials are good candidates to enter the world of hybrid composites for aerospace [1,2].

**Author Contributions:** All the researchers collaborated for the main activities of the study, main contributions may be identified as follows: Conceptualization, F.Q.; methodology, D.B. and A.P.; validation, A.P. and N.G.; formal analysis, F.Q. and L.S.; investigation, A.P., N.G. and D.B.; resources, L.S.; data curation, A.P. and F.Q.; writing-original draft preparation, A.P., F.Q. and D.B.; writing—review and editing, D.B. and N.G.; visualization, A.P.; supervision, L.S.; project administration, F.Q. and L.S. All authors have read and agreed to the published version of the manuscript.

**Funding:** This research received no external funding.

**Institutional Review Board Statement:** Not applicable.

**Informed Consent Statement:** Informed consent was obtained from all subjects involved in the study.

**Data Availability Statement:** Not applicable.

**Acknowledgments:** The authors are grateful to Fabrizio Betti for the support in the experimentation.

**Conflicts of Interest:** The authors declare no conflict of interest.

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
