# Peer review of "Damping Behavior of Hybrid Composite Structures by Aeronautical Technologies"

_applsci, doi:10.3390/app12157932_

Round 1

Reviewer 1 Report

check line 69, ref [22] is NOT dealing with the interaction of glass and carbon fibres, but with the interaction of carbon an steel fibres

2. Materials and Methods: areal weight (g/m^2) and cured ply thickness (mm) of the prepreg are missing

2.2 autoclave heating and cooling rates are missing

Figure 3: scales are not readable, contrast of images is extremely poor

2.3: conditioning of samples = ? tests at room temperature (?) (results would certainly be different at different ambient conditions....)

Author Response

Q1) check line 69, ref [22] is NOT dealing with the interaction of glass and carbon fibres, but with the interaction of carbon an steel fibres

A1) Thanks to the reviewer for checking this missing, now the authors had changed the line as follow: “The hybridization of thermoplastic matrices was also investigated showing positive interactions on static and dynamic behavior both in the case of glass and carbon fibers with polypropylene [21] as well as for carbon fibers and metallic braided wire mesh with poly-ether-ether-ketone [22].”

Q2) 2. Materials and Methods: areal weight (g/m^2) and cured ply thickness (mm) of the prepreg are missing

A2) We are grateful to the reviewer for the suggestion about missing data, we added few lines in section 2.2 as follow “The measured cured ply prepreg thickness is 0.24 mm and the measured areal weight of the prepreg is 335 g/m2.“

Q3) 2.2 autoclave heating and cooling rates are missing

A3) Thanks to the reviewer, we added the missing data in section 2.2 as follow: ”with a heating and cooling rate of 2 °C/min".

Q4) Figure 3: scales are not readable, contrast of images is extremely poor

A4) We agree with the reviewer and we enlarged the scale bars and improved the image’s contrast so as to make the different layers visible.

Q5) 2.3: conditioning of samples = ? tests at room temperature (?) (results would certainly be different at different ambient conditions....)

A5) Thanks to the reviewer for the suggestion, the missing data about conditioning of the samples were added. The authors inserted the ambient conditions for all the tests performed.

Reviewer 2 Report

Dear Authors,

I read your contribution with interest and attention. I can surely say that there are several remarkable points for reflection and, in general, that the manuscript is valid. At the same time, I have to say that there are several issues that require correction or explanation.

Kind regards

1. Please avoid citation concentrations. The introduction is expected to have an extensive literature review followed by an in-depth and critical analysis of reffered sources. Authors should avoid reference overkill/run-on.

2. The novelty of the work should be clearly indicated in the introduction.

3. Please provide the description of the manuscript structure. Provide a brief description of the content of each section (see 10.3390/ma13071630).

4. Please double check the Figures numbering.

5. Please add some Figures with testing setup details, both for static and dynamic analysis. 

6. Please explain in detail how the damping was determined. How the loss factor from hysteretic damping is related to a loss factor determined e.g. based on a half power method.

7. Please add a discussion on how the proposed solution looks compared to others. Please indicate the advantages and limitations comparing to similar studies.

Author Response

Q1) Please avoid citation concentrations. The introduction is expected to have an extensive literature review followed by an in-depth and critical analysis of reffered sources. Authors should avoid reference overkill/run-on.

A1) Thanks to reviewer for the suggestion, the authors agree with this comment and the references have been discussed more in depth. Now the new introduction section has several improvements.

Q2). The novelty of the work should be clearly indicated in the introduction.

A2) Thanks to the reviewer, in this regard this sentence has been added to clarify this aspect: “For the first time, hybrid composite laminates have been manufactured by interposing uncured rubber interlayers between CFR thermosetting prepregs during the lamination procedure. Another novel aspect is that hybrid composites have been produced by using aeronautical materials and moulding technologies at the top of the state-of-the-art”.

Q3) Please provide the description of the manuscript structure. Provide a brief description of the content of each section (see 10.3390/ma13071630).

A3) Thanks to the reviewer for the suggestion. The description of the manuscript has been added at the end of the introduction section as follow: “The article is organized in 5 sections as follow: in section 2 materials properties, samples architectures and manufacturing as well as the whole experimental procedure are reported. The results of both static and dynamic tests are shown in section 3 and then fully discussed in section 4. The final section includes a summary of the paper main results and the conclusions”

Q4) Please double check the Figures numbering.

A4) Thank you, we double checked the Figures numbering.

Q5) Please add some Figures with testing setup details, both for static and dynamic analysis. 

A5) Thanks to the reviewer for the kind suggestion, in order to explain better the two different tests, a new Figure 4 was added.

Q6) Please explain in detail how the damping was determined. How the loss factor from hysteretic damping is related to a loss factor determined e.g. based on a half power method.

A6) Thanks to the reviewer for this kind suggestion. For this point, this explanation has been added: “The damping behaviour has been measured by DMA, free oscillation and cycling testing. Results are expressed in terms of loss factor (which is close to the half of the material damping coefficient), energy spent per loading-unloading cycle, and acceleration reduction under free oscillation. These data provide a description of the material damping capabilities under a wide spectrum, from small to large displacements. They are not dependent one from the other but a correlation may exist. Generally, an improvement of the damping behaviour of a structure leads to a general increase of all these parameters, but with different levels of extent”.

Q7). Please add a discussion on how the proposed solution looks compared to others. Please indicate the advantages and limitations comparing to similar studies.

A7) As suggested by the reviewer in order to discuss this point, this part has been added: “The proposed architectures and material combinations are quite new in the scientific database; therefore, a direct comparison with other similar studies is difficult to do. Nevertheless, the known advantage in terms of mixing the elastic and damping properties of the raw materials has been observed, even if final products exceed in the damping performances thus reducing strongly the elastic part”.

Round 2

Reviewer 2 Report

Dear Authors, 

thank You for Yours reply. Now I can recommend the manuscript for publication. 

Perhaps the only additional suggestion is to include the impact test of a free-free samples in future studies. This will allow to eliminate the influence of the fixing on the dynamic properties of the samples.

Kind regards

Author Response

This suggestion has been included in the manuscript by the sentence: “In the future, new architectures will be tested by taking into consideration also different configurations, such as free samples under impact probes”.